# Serotonin 1A Receptor Pharmacotherapy and Neuroplasticity in Spinal Cord Injury

**DOI:** 10.3390/ph15040460

**Published:** 2022-04-11

**Authors:** Afaf Bajjig, Florence Cayetanot, J. Andrew Taylor, Laurence Bodineau, Isabelle Vivodtzev

**Affiliations:** 1Sorbonne Université, Inserm, UMR_S1158 Neurophysiologie Respiratoire Expérimentale et Clinique, F-75005 Paris, France; afaf.bajjig@sorbonne-universite.fr (A.B.); florence.cayetanot@sorbonne-universite.fr (F.C.); laurence.bodineau@sorbonne-universite.fr (L.B.); 2Department of Physical Medicine and Rehabilitation, Harvard Medical School, Cardiovascular Research Laboratory, Spaulding Rehabilitation Hospital, Cambridge, MA 02139, USA; jandrew_taylor@hms.harvard.edu

**Keywords:** serotoninergic pharmacology, spinal cord injury, neuromodulation, locomotion, respiration

## Abstract

Spinal cord injury is associated with damage in descending and ascending pathways between brainstem/cortex and spinal neurons, leading to loss in sensory-motor functions. This leads not only to locomotor reduction but also to important respiratory impairments, both reducing cardiorespiratory engagement, and increasing cardiovascular risk and mortality. Moreover, individuals with high-level injuries suffer from sleep-disordered breathing in a greater proportion than the general population. Although no current treatments exist to restore motor function in spinal cord injury (SCI), serotoninergic (5-HT) 1A receptor agonists appear as pharmacologic neuromodulators that could be important players in inducing functional improvements by increasing the activation of spared motoneurons. Indeed, single therapies of serotoninergic 1A (5-HT_1A)_ agonists allow for acute and temporary recovery of locomotor function. Moreover, the 5-HT_1A_ agonist could be even more promising when combined with other pharmacotherapies, exercise training, and/or spinal stimulation, rather than administered alone. In this review, we discuss previous and emerging evidence showing the value of the 5HT_1A_ receptor agonist therapies for motor and respiratory limitations in SCI. Moreover, we provide mechanistic hypotheses and clinical impact for the potential benefit of 5-HT_1A_ agonist pharmacology in inducing neuroplasticity and improving locomotor and respiratory functions in SCI.

## 1. Introduction

Spinal cord injury (SCI) affects more than 2.5 million people worldwide and is one of the most devastating conditions that an individual can sustain, leading to partial or complete loss of motor and sensory function below the injury level. Disruption of spinal cord integrity alters descending supraspinal pathways affecting motor drive, as well as ascending pathways affecting sensitivity. When injury is high, SCI results not only in motor impairment but also in marked respiratory dysfunction [1] due to alteration of the bulbo-spinal and cortico-spinal pathways. Furthermore, damage in descending and ascending pathways between brainstem/cortex and locomotor or respiratory neurons drastically reduces muscle efferents and afferents [2]. Indeed, central respiratory drive only occurs over physiologically active connections and not over less active or silent connections [3]. Hence, the damaged ascending pathways impair feedback to brainstem, cerebellum, and subcortical areas, resulting in dysregulation of the spinal network, impacting at both locomotor and respiratory levels. 

### 1.1. Locomotor Impact

One main consequence of SCI is the reduction in sensorimotor function, which results from a failure in functional innervation following injury despite spontaneous axonal regrowth [4,5]. Moreover, muscle disuse adds to the deficit and reduces the potential for recovery from neuro-regenerative therapy [4]. Despite major progress over the past 30 years, targeting neurobiological mechanisms underlying neuroregenerative recovery, restoration of volitional locomotion and/or arm-hand function remain top priorities and have been targeted for noninvasive therapies to improve function after SCI [5]. This is even more true since individuals with SCI now live longer, and lack of functional improvement can substantially increase rates of severe complications. Indeed, a major indirect consequence of reduced locomotor function is the lack of cardiorespiratory engagement, increasing cardiovascular risk and mortality [6]. In fact, SCI is associated with a 3 times greater risk for cardiovascular disease compared with the general population due to the loss of metabolically active tissue, reduced capacity to routinely engage in aerobic exercise, and/or inability to achieve and maintain high levels of cardiorespiratory fitness [7,8].

### 1.2. Respiratory Impact

In addition to locomotor deficits, high-level spinal cord injury also reduces descending supraspinal respiratory pathways proportional to the level of injury. As a result, complete lesions at the level of the third cervical vertebrae (C3) or above induce diaphragm paralysis, making autonomous breathing impossible. The highest-level injuries render survivors ventilator dependent, dramatically compromising quality of life [9] and increasing mortality [10]. After incomplete lesions or lesions between C3 and C5, the diaphragm is partially disconnected from the supraspinal respiratory network and ventilation is compromised. Moreover, extra-diaphragmatic respiratory muscles with thoracic innervation are also impacted by cervical SCI and cannot compensate for reduced diaphragmatic contraction. This reduction in function inexorably leads to respiratory muscle wasting and weakness. For example, even short periods of mechanical ventilation that suppress diaphragmatic work have a drastic impact on diaphragmatic muscle structure in sheep [11], leading to muscle fiber atrophy after only two days. Hence, cervical and upper thoracic injuries lead to reduced activation of respiratory motoneurons and thus at least partial respiratory failure. 

Furthermore, although lower cervical injuries (<C5) spare the upper limbs, the reduction in respiratory capacity with locomotor limitations worsens morbidity by limiting intensity during exercise-based rehabilitation, an important parameter for cardioprotection with rehabilitation [8]. Indeed, limited respiratory function in those with high-level injuries may be a limiting factor to whole-body exercise via functional electrical stimulation -FES- [12] which allows much greater exercise intensities, such that muscle metabolic demand exceeds ventilatory capacity [13]. We recently found that if respiratory function is improved via ventilatory support during exercise, this limitation can be overcome, and cardioprotective levels of exercise intensity can be achieved [14]. 

Lastly, the prevalence of sleep-disordered breathing in SCI is markedly greater than in the general population, estimated at 27–77% as compared to 2–4%. This likely derives from lower effective regulation of (CO_2_) for a given alveolar ventilation due to the inability of the supraspinal respiratory network to increase the activity of spinal respiratory motoneurons [15].

Currently, there are no treatments to restore motor function in SCI. There is an urgent need to develop accessible interventions for both acute and chronic SCI that induce neuroplastic changes and reactivation of the bidirectional neural circuitry between brain and muscle, not only to induce functional recovery but also to improve respiratory and cardiorespiratory functions.

### 1.3. Serotonin 1A-R agonists and Neuromodulation

Among recent approaches to increasing functional recovery after SCI, pharmacologic neuromodulation is an important player for neuroplastic changes to induce functional improvement. Most individuals with SCI, even those who are “clinically” complete, have only partial anatomic disruption of the spinal cord. Pharmacologic neuromodulation, therefore, presents an extraordinary opportunity to increase the activation of spared motoneurons. Serotoninergic (5-HT) agonist administration could boost functional improvement. Indeed, 5-HT pathways powerfully modulate spinal motor control via intrasynaptic and extrasynaptic receptors [16]. There are multiple effects of exogenous 5-HT on spinal rhythmic activities and motoneuron excitability, suggesting that it could play an important role in modulating the activity of spinal networks involved in vertebrate locomotion (see Perrier et al. for review [16]). SCI is associated with dysregulation of spinal 5-HT receptors [17] and depletion of 5-HT in the spinal locomotor neural network [18], and each contributes to loss of locomotor function below the lesion level. However, there is subsequent upregulation in the expression and sensitivity of specific 5-HT receptors (e.g., 5-HT_1A_R and 5-HT_2A_R) below the injury site in regions associated with hindlimb function [19]. As a result, an important part of locomotor reorganization after SCI involves 5-HT pathways that could be activated by 5-HT agonists [20] (Figure 1). 

Moreover, it is increasingly understood that serotonin (5-HT) influences the central respiratory drive and spinal respiratory motoneurons. The consensus suggests that 5-HT released by neurons of the medullary raphe nuclei stimulates the brainstem respiratory network, although the respective roles for receptor subtypes remain unclear [21]. Hence, an important part of respiratory reorganization after cervical injury involves serotonergic pathways that can be activated by 5-HT agonists too [20] (Figure 1). Among them, 5-HT_1A_ receptor agonists could be of particular interest in SCI, given growing evidence indicating the extensive value of 5-HT_1A_ pharmacotherapy beyond the well-known anxiolytic effects. For example, respiratory-related neural activity in the brainstem and spinal cord is significantly reduced in rats [22] and administration of a 5-HT_1A_ agonist both systemically and locally (on the dorsal horn of the spinal cord) increases phrenic motor output from the injured spinal cord after both acute and chronic injuries [23]. 

In this unique review, we hence discuss previous and emerging evidence showing the value of 5HT_1A_ receptor agonist therapy for motor and respiratory limitations in SCI. More precisely, the purpose of our study is to provide a robust insight into the work delivered in the last two decades on how 5HT_1A_ receptor agonists can be beneficial in clinical cases outside of their common use in psychiatry as anxiolytics. Particularly, studies suggest that 5HT_1A_ receptor agonist therapies led to improvement in both locomotor and respiratory functions, and that this effect was amplified when 5HT_1A_ receptor agonist treatments were combined with other drugs, electrical stimulation of the spinal cord, or exercise training (rehabilitation). This is the first study providing an exhaustive overview of the literature on the effect of 5HT_1A_ on locomotor/respiratory neuroplasticity in SCI, not only providing new insight on the use of this pharmacotherapy in the rehabilitation of individuals with SCI for clinicians, but also future directions for scientists aiming at developing research in neuroplasticity in SCI.

Herein, we searched for published studies investigating Serotonin 1A receptor pharmacotherapy and neuroplasticity in spinal cord injury. PubMed and Web of Science databases were screened using the following keywords and MeSH terms: (spinal cord injury) AND (Serotonin 1A receptor) AND (locomotor) OR (respiration). Original studies in animals or in humans that met the following criteria were included: (i) study design: drug alone or combined; within or between groups comparison, non-randomized between-groups studies, randomized controlled studies, cross-sectional studies, case, and cohort studies (for humans); (ii) subjects: SCI models including total or partial transection, contusion, decerebration/individuals with spinal cord injury; and (iii) outcomes: any outcomes related to locomotor or respiratory function. Non–English language articles, review articles, and congress abstracts were excluded. 

## 2. Locomotor Effects of 5-HT_1A_ Agonists Pharmacology in SCI

Our search identified 18 studies that reported locomotor outcomes using 5-HT_1A_ receptor agonists in animal models of SCI or in humans. These studies were homogenous in finding positive changes in locomotor function, although with different designs and outcomes (see Table 1). 

### 2.1. Isolated Effects of 5-HT_1A_ Agonist Pharmacology

All but one study used pre-clinical mammal models of SCI and most employed intraperitoneal (i.p.) injection. The very first observation of benefit from a 5-HT_1A_ receptor agonist on locomotor function was in isolated rat dorsal columns. Saruhashi et al. found greater recovery of mean evoked potential amplitudes induced by supramaximal constant current electrical stimuli with Tandospirone (a 5-HT_1A_ agonist) perfusion compared with Ringer’s solution [24]. This was the first evidence of improved recovery of spinal nerve potential amplitudes in a rat spinal cord injury model. A few years later, Landry et al. reported that a single dose (1 mg⁄kg, i.p.) of 8-hydroxy-2-(di-Npropylamino)-tetralin (8–OH-DPAT), a potent and selective 5-HT_1A_ agonist, induced hindlimb movements within 15 min of administration in mice with a spinal cord transected at the low-thoracic level (T9) [25]. This effect was inhibited when the paraplegic mice were pretreated with a selective 5-HT_1A_ receptor antagonist. This result was corroborated by a second study by the same group, who showed that an even lower dose of 8-OH-DPAT (0.5 mg/kg, ip) could acutely elicit some locomotor-like movements (defined as flexion followed by extension or vice versa occurring in both hindlimbs) in completely spinal cord transected mice [26]. Overall, these results provide the first evidence of the benefit of 5-HT_1A_ agonists on locomotion in low-thoracic-transected mice. 

Despite these encouraging results, no other studies were performed over the next 10 years. More recently, Jeffrey-Gautier et al. [27] reported that treatment with systemic 5-HT_1A_ agonists (buspirone: 8mg/kg i.p.) can acutely initiate locomotor function in T8 transected mice [27]. Indeed, the authors found enhanced stepping, paw placement, and lower limb joint excursions during the step cycle as early as 2 days after complete transection. Moreover, it is likely that locomotor improvements are associated with improved motoneuron reflex excitability, as suggested by Develle and Leblond. These authors showed that the same single dose of Buspirone could increase frequency-dependent depression of the H-reflex as compared with saline injection [28] in mice after T8 complete spinal lesion. Moreover, Jin et al. [29] showed that the same 5-HT_1A_ agonist can improve forelimb locomotion by increasing reaching and grasping accuracy as early as 2 weeks post C4 injury in adult female rats [29] using a lower dose of Buspirone (1–2 mg/kg/day, i.p.). Of note, this finding was confirmed by the work of Ahmed et al., 2021 who found the highest reaching scores and grip strength in the lowest-dose group rats (1.5 mg/kg/day, i.p.) compared to 2 other dose groups (2.5 and 3.5 mg/kg/day), indicating that forelimb recovery with Buspirone is dose-dependent after cervical spinal cord injury with the best performance occurring at the lowest dose.

### 2.2. Combinatorial Effects of 5-HT_1A_ Agonist with Other Pharmacotherapies

Although single therapies of 5-HT_1A_ agonists allow for acute and temporary recovery of locomotor function, a number of studies strongly suggest that combining 5-HT pharmacotherapy with other rehabilitative approaches could be more promising than the drug alone. Indeed, 5-HT_1A_ agonist may have greater effects or may induce long-lasting effects when combined with pharmacotherapy, spinal stimulation, or exercise rather than administered alone. For example, Slawinska et al. showed that when combined with quipazine (a dopamine agonist), a single dose of 8-OH-DPAT (0.2–0.4 mg/kg, ip) was able to improve locomotor activity in both horizontal and upright posture in complete T9 transected rat [30]. Moreover, Lapointe et al. showed that when combined with a SKF-81297, a dopamine D1/5 receptor agonist, 8-OH-DPAT (0.5 mg/kg, ip) can elicit stepping movements in complete paraplegic animals (T9 transection) [26]. In another study, Guertin et al. explored the effect of a combination therapy consisting of 1.5 mg/kg buspirone, 1.5 mg/kg apomorphine and 50 mg/kg Levodopa, combined with 12.5 mg/kg of benserazide, a decarboxylase inhibitor, to increase its central bioavailability. In this proof of concept study, the Canadian group showed that when administered subcutaneously 4 times per week for 1 month, this combined therapy can induce locomotor activity as assessed by hindlimb stepping and weight-bearing on a motorized treadmill in the T9 transected mice [31]. The authors suggested that this drug combination would also activate central pattern generators in chronic SCI individuals [32]. In fact, the safety and preliminary efficacy of this therapy have been tested in individuals with SCI by the same group of authors in 45 subjects with complete or complete/sensory incomplete SCI (ASIA A or B) for at least 3 months [33]. The authors found rare signs of safety concerns (mild nausea in 3 cases) at a low dose of SpinalonTM (10–35 mg buspirone/100–350 mg levodopa/25–85 mg carbidopa) and treated subjects had improved leg muscle EMG with locomotor-like characteristics [33] (Table 1). 

### 2.3. Combinatorial Effects of 5-HT_1A_ Agonist with Rehabilitation

Ung et al. combined Buspirone with Carbidopa (an aromatic L-amino acid decarboxylase inhibitor and Clenbuterol, an agonist at the beta-2 adrenergic receptor) as well as exercise training. This study was the first to investigate the synergistic effects of exercise with 5-HT pharmacotherapy on locomotor function in a mice model of SCI [34]. The treatment led to increased locomotor movement associated with increased lower limb muscle fiber size in T9 transected mice vs. untreated and untrained animals. However, although treatment combination, particularly SpinalonTM (Buspirone/Levodopa/Carbidopa) may have some promising positive effect on locomotion in SCI, it was not possible to discriminate the specific benefit of the 5-HT_1A_ agonist in this pharmacological combination. 

In a more recent study, Ganzer et al. assessed locomotor recovery and dendritic density (1 and 9 weeks) after SCI following therapeutic interventions, including a combination of 5-HT pharmacotherapy (5-HT_2A_R, 5-HT_1A_R agonists), bike therapy, or both [35]. They found that an acute dose of 5-HT_2A_R and 5-HT_1A_R agonist improved locomotor behavior when administrated late (week 9 post-injury) after SCI compared to early (week 1 post-injury). Although this is the first study showing evidence that 5-HT pharmacotherapy can mediate recovery following SCI partly by restoring spinal dendritic density, the locomotor improvement was not sufficient to allow animals to achieve weight-supported stepping, and single therapies did not normalize the loss of dendritic density after SCI.

The potential synergistic effect of 5-HT_1A_ agonist with rehabilitation was suggested by two studies, both in mice and in humans. Jeffrey-Gautier et al. showed that Buspirone can induce long-lasting improvements of paw positioning at contact and paw drag in mice when combined with training sessions on the treadmill [27]. Moreover, this occurred after a dual lesion paradigm suppressed the supraspinal pathway above the lesion level, suggesting that buspirone exerts considerable acute facilitation of spinally mediated locomotion. In addition, our recent findings suggest that buspirone at clinically standard doses (30 mg/day) may enhance gains in aerobic capacity during whole-body FES exercise training in 11 patients with high-level SCI. This result may be related to an improvement at the locomotor level, but we also found a strong association with improvement of ventilatory capacity during exercise. For this reason, we will provide more details about this study in the respiratory section [36]. 

Of note, in a retrospective analysis, Morgan and Solinsky showed that there was no difference in functional scores following acute traumatic SCI when buspirone is initiated in an uncontrolled fashion during acute inpatient rehabilitation [37]. However, in individuals with clinically complete SCI, findings suggest possible increased rates of 1-year conversion to incomplete injury, and this was more pronounced when rehabilitation was performed with more intense forms of exercise training such as FES rowing.

### 2.4. Combinatorial Effects of 5-HT_1A_ Agonist with Spinal Cord Stimulation

The potential benefit of 5HT_1A_ receptor agonists to enhance locomotor recovery after SCI has also been studied in combination with electrical transcutaneous spinal cord stimulation. All studies were conducted in humans across 12 participants. Moshonkina et al. showed that buspirone (taken orally twice a day for 18 days in a dose-escalation manner) induced greater improvement in leg muscle activation, tone, and strength after a four-week course of spinal cord electrical stimulation in patients with thoracic SCI (n = 5) [38]. Moreover, according to the Medical Research Council and reflex scores, they found greater positive shifts in the pain sensitivity index and force index over 1 year in the Buspirone group. In fact, this combination of treatments (Buspirone + spinal stimulation) may enhance stepping motion, as well as coordination patterns of the lower limb muscles on stepping-like movement using an exoskeleton (a robotic device assisting stepping), as shown by Gad et al. in a completely paralyzed patient (>4 year) [39]. Lastly, the combination of spinal stimulation and buspirone (20 mg/day for 15 days) was assessed on upper limb motor function in 6 individuals with chronic (>1 year) motor complete cervical injury (C5 or above). Freyvert et al. found almost doubled hand strength with the combination compared to spinal stimulation alone [40]. Taken together, these studies support the hypothesis of the synergistic effect of buspirone and spinal stimulation on locomotion in SCI. Of note, spinal stimulation and rehabilitation are two different programs that trigger different effects. Indeed, spinal cord stimulation is a local procedure consisting of an electrical current application via cutaneous electrodes on the spinal cord to activate intraspinal neural networks [41], while during rehabilitation, individuals with SCI perform exercise with their non-paralyzed muscles and/or their paralyzed muscles using devices such as electrical stimulation, weight-bearing system and/ or exoskeleton, therefore triggering neuromuscular or muscular effects [6].

## 3. Respiratory Effects of 5-HT_1A_ Agonist Pharmacology in SCI 

The effect of 5-HT_1A_ agonist on respiratory function in SCI was first investigated by Teng et al. in rats after either T8 contusion [42] and Choi et al. in rats after C5 hemicontusion [43] (Table 2). This model of contusion (rather than section) leading to spared motoneurons and not death of all the neurons is closer to human traumatism and clinical settings [44]. The authors found that both 8-OH-DPAT and buspirone (a 5HT_1A_ partial agonist) improved respiration post-SCI, while pretreatment of animals with p-MPPI, a 5-HT_1A_ selective antagonist, abolished the beneficial impact of either drug [43]. This study also suggests that Buspirone acts on respiratory function via activation of 5-HT_1A_ receptors. Of note, despite low thoracic injury, Teng et al. found an altered respiratory function with decreased tidal volume (V_T_) (−27%), increased breathing frequency (B*f*) (+33%), and lower ventilatory response to 7% CO2 (−32%) after complete T8 contusion. These abnormalities were reversed for up to 4h (with a peak of efficiency at 20 min post administration) by administration of a 5-HT_1A_ agonist 24 h after injury [42]. Based on these results, the same team performed a similar study in a C5 hemi-contusion model. The hemi-contusion at a higher level led to similar lower V_T_, increased B*f_,_* and lower ventilatory response to 7% CO_2_. In this study, Buspirone or 8-OH-DPAT administered intraperitoneally did not change breathing in ambient air but restored, in part, the ventilatory response to hypercapnia [43] at both 2- and 4-weeks post-injury, suggesting that this effect may also be present in the chronic stage of SCI.

The potential benefit of 5-HT_1A_ agonists on breathing was supported by another team in a model of C2 hemisection. Zimmer et al. found that both spinal and systemic application of 8-OH-DPAT increased bilateral phrenic motor output and elicited latent crossed phrenic activity in rats both acutely (24–48 h) and chronically (16 weeks) injured. This study suggests that respiratory improvements are associated with changes in diaphragmatic activity, even in chronic stages of injury. 

Together, these studies suggest that systemic 5-HT_1A_ agonists can temporarily restore respiratory function in rats after SCI section. However, only single doses of 5-HT_1A_ agonists have been used, and longer-term effects are still unknown. Moreover, since 2006, no further ork has explored the effects of Buspirone on respiratory function in SCI. However, based on prior observations, Vivodtzev et al. recently investigated the effect of chronic Buspirone administration combined with whole-body exercise training over 3 months in individuals with high level SCI (>T3) [36]. This retrospective assessment suggested a positive, synergistic effect of Buspirone and whole-body exercise on aerobic capacity and respiratory function in patients during the sub-acute stage of high-level SCI. A strong relationship of changes in aerobic capacity to ventilatory capacity was found, as well as to functional vital capacity and forced expiratory volume, suggesting that improved breathing may allow for improved aerobic exercise capacity in these individuals. This study suggests the value of combined pharmacotherapy and exercise for both improved respiratory function and improved exercise capacity in SCI.

Lastly, Buspirone may have important effects on nocturnal ventilatory instability in individuals with chronic SCI. Indeed, Maresh et al. recently investigated the effect of a chronic oral administration of Buspirone for 13 days in patients with high level SCI (>T3) and sleep disordered breathing [45]. In this study, Buspirone triggered a widening of CO_2_ reserve and a decrease in hypocapnic central sleep apnea. This is the first study suggesting a positive effect of 5-HT_1A_ agonists on sleep-disordered breathing in SCI.

## 4. Mechanistic Hypotheses 

Although studies suggest the benefit of buspirone on both locomotor and respiratory recovery after SCI, the mechanisms of action of 5-HT_1A_ receptor agonists on neuromodulation in SCI are not completely understood. Both excitatory and inhibitory inputs to locomotor and respiratory motoneurons may be induced by 5-HT_1A_ receptor activation, depending on the target and its location [46]. Nevertheless, a direct effect of 5-HT_1A_ receptor on breathing has been suggested by Dr Teng et al. The primary mechanism appears to be drug-induced 5-HT_1A_ receptor activation increasing the excitability of the ventral respiratory motoneurons that survived the spinal cord injury. This would occur via inhibition of inhibitor interneurons in the ventral horn of the spinal cord. The fact that 5-HT_1A_ agonists potentiate the effects of spinal stimulation [40] suggests that 5-HT_1A_ agonist may enhance the level of excitability of the motor network of the spinal cord (including Central Pattern Generator) mediating motor function [40]. As previously reported in cervical spinal stimulation, it may impact spinal interneuronal pathways that activate motoneurons [39]. As a result, when combined, buspirone and spinal stimulation would lead to a co-activation of specific targets that may induce greater cellular activity within the spinal network, leading to the generation of miniature excitatory potentials and thus shifting the spinal motor network excitability closer to the motor threshold [31,39]. This would explain why buspirone would be more effective when combined with other approaches to neuromodulation and would suggest that activation of spinal interneurons or inhibition of inhibitory interneurons is a potential mechanism for the excitatory effect of 5-HT_1A_ agonist administration.

The dose may condition the effect of 5-HT_1A_ agonist. Indeed, 5-HT neurons in the medullary raphe nuclei fire regularly at frequencies tightly correlated with motor activities [47] such that the amount of 5-HT released in the spinal cord varies in parallel with movement intensity [16]. As demonstrated by Ahmed et al., a low level of Buspirone (1.5 mg/kg) is more effective than a higher level (2.5 and 3.5 mg/kg) for induction of locomotor benefit. Although a dose of 8 mg/kg Buspirone also led to functional improvement in a similar model of SCI, the optimal dosage and underlying mechanisms are yet to be defined. Nevertheless, 1.5 mg/kg of Buspirone seems to be the most common dose used across studies (for both single or repeat doses).

Another potential mechanism of 5-HT_1A_ receptor activation would be to increase the presynaptic transmission of proprioceptive sensory inputs. Indeed, 5-HT_1A_ receptors seem to be involved in specific spinal reflex patterns in cats [48]. Since the densest population of 5-HT_1A_ receptors in the spinal cord is in the dorsal laminae of the spinal cord, this might suggest the role of this receptor in processing sensory inputs. Thus, buspirone might act in the dorsal horn to restore locomotor or respiratory function to normal by acting on neural pathways in the dorsal spinal cord. 

As an example, our recent findings suggest that Buspirone potentiates the effects of whole-body exercise on respiratory function, perhaps by enhancing respiratory drive via pathways at the spinal levels. Drug-induced 5HT-_1A_ receptor activation may indeed cause an increased excitability of ventral motoneurons that survived the spinal cord injury, as previously shown in a rat model of cervical SCI [42]. Therefore, when coupled with exercise training, Buspirone may have resulted in greater volume and flow changes with exercise. In addition, intercostal and abdominal muscle afferents send their inputs to the dorsal horn, which has the densest population of 5-HT_1A_ receptors in the spinal cord [49] and can modulate tidal volume (V_T_) [50]. Hence, Buspirone might act on neural pathways of the dorsal horn by increasing the impact of facilitating inputs from muscle afferents on phrenic motoneurons. This would explain why the combination of exercise training, *which would greatly increase afferent activity*, and Buspirone, *which would increase motoneuron excitability and inputs from* intercostal and abdominal muscle, is more effective than exercise training alone to improve ventilatory capacity (Figure 2). 

Lastly, the fact that buspirone improved breathing instability in patients with SCI and sleep disordered breathing suggest that 5-HT_1A_ receptors may play a role. Marseh et al. suggested that buspirone lowers susceptibility to hypocapnic central apnea by blunting chemosensitivity (and reducing loop gain) and increasing ventilation [45]. Whether 5-HT_1A_ receptors may reduce other forms of sleep apnea deserves to be investigated, and the mechanisms of action of Buspirone on sleep apnea need to be studied in more detail. 

## 5. Clinical Impact

When administered as a monotherapy, 5-HT_1A_ agonists showed acute and temporary recovery of locomotor or respiratory function [27,43]. Although these studies have mainly been in animals, drug dosages and administration mode reproduce current practices in humans, allowing for faster transition to humans. These studies provide key evidence to prompt further investigation into this potential therapeutic strategy. Moreover, a clinical dose of Buspirone (30 mg/day) may reduce the susceptibility to hypocapnic central sleep apnea in SCI [45], a result which is corroborated by similar findings in case reports of other respiratory disorders in multiple sclerosis [53], and Rett syndrome [54], confirming a potential therapeutic effect of Buspirone alone on respiratory function and particularly in sleep-disordered breathing in SCI. 

However, single therapies of 5-HT_1A_ agonists appear to be insufficient to compensate for motoneuronal loss after SCI or to induce long-lasting effects [20]. In animals, respiratory effects lasted 4h [43] and repetitive administration of buspirone has not been investigated on diurnal ventilation in SCI. In addition, despite grasping improvements, Buspirone alone failed to improve quadrupedal locomotion in C4-injured rats [55] and 5-HT_1A_ agonist alone cannot compensate for the loss of skeletal muscle and therefore has limited effect on locomotion. 

On the contrary, as suggested in this review, further gains in recovery may only be expected when 5-HT_1A_ pharmacology is combined with other approaches that aim to induce neuromodulation. Indeed, a pharmacological combination would be able to activate locomotion in chronic SCI individuals. Moreover, 5-HT_1A_ treatment enables long-lasting gains in locomotion in mice [27] and human [39], improving muscle coordination and upper limb strength when combined with appropriate rehabilitation in humans [27,34,40]. Furthermore, studies have found a synergistic effect of buspirone and spinal stimulation on locomotion in SCI, enhancing stepping motion in the exoskeleton and muscle strength. The authors suggest that this drug combination would also activate the central pattern locomotion in chronic SCI individuals. Of note, it is likely that when associated with spinal stimulation or with rehabilitation, 5-HT_1A_ agonist pharmacotherapy would increase the shift from complete to incomplete injury, representing a huge change in patient quality of life, with potential higher autonomy [37,38]. 

## 6. Perspectives and Future Directions

Although 5-HT_1A_ agonist pharmacotherapy seems very promising in SCI, there is little understanding of how these treatments may individually or synergistically promote recovery [35]. Very few studies have investigated the mechanisms by which 5-HT_1A_ receptor agonists improve locomotor and respiratory spinal networks. Future work needs to investigate, in a pre-clinical approach mimicking clinical implementation, whether 5-HT_1A_ agonists do increase nerve sprouting, motoneuron excitability, and/or neuroplasticity to induce functional recovery after SCI. For instance, there are few studies on the effect of 5HT_1A_ agonists on contusion models of cervical SCI, although they are more clinically relevant compared to section models, since they lead to spared motoneurons and not death of all the neurons. It is likely that the hemi-contusion model at the C3/C4 level could be well adapted to investigate respiratory and locomotor function, as it would theoretically impact ipsilateral phrenic motoneurons and upper limb motoneurons in addition to lower limb and trunk muscles. Moreover, more mouse models are needed, since there is currently almost no alternative to mice for studying the roles of specific genes after SCI. Further projects could, for example, explore and confirm the effect on genetically engineered lines (such as 5HT_1A-_R KO mice [56]).

## Figures and Tables

**Figure 1 pharmaceuticals-15-00460-f001:**
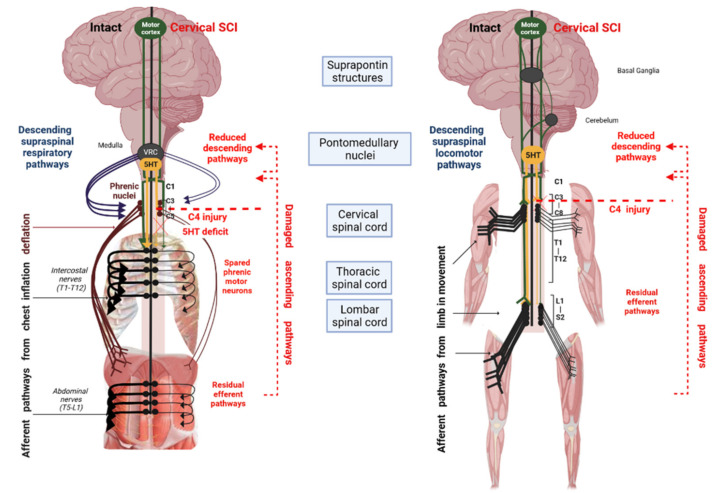
Schematic of the effects of cervical SCI on spinal pathways (**left**: respiratory and **right**: locomotor) and the potential role of serotonin (5-HT). After cervical SCI, the interruption of supraspinal descending pathways leads to a reduced effect of 5-HT on spinal motoneuron excitability.

**Figure 2 pharmaceuticals-15-00460-f002:**
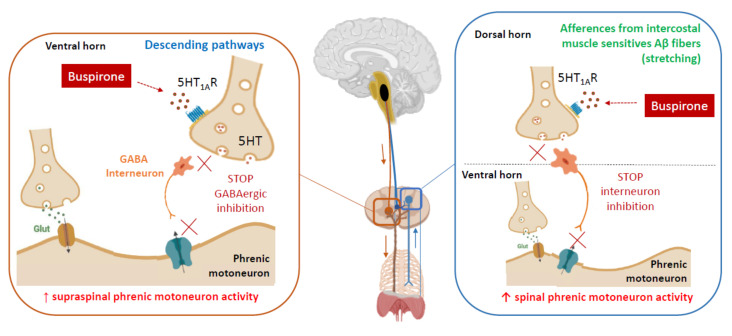
Expected mechanisms of action of Buspirone on respiratory network. Buspirone may directly increase the excitability of spared phrenic motoneurons by inhibiting an inhibitor interneuron in the ventral horn of the spinal cord. Alternatively, or in addition, it may enhance presynaptic transmission of proprioceptive sensory inputs from intercostal and abdominal muscles by inhibiting an inhibitor interneuron in the dorsal horn of the spinal cord [42,51,52].

**Table 1 pharmaceuticals-15-00460-t001:** Locomotor effects of 5-HT_1A_ in animal and humans.

Authors	Species/Type of SCI	Drug and Dose	Combined with	Method	Main Outcome Measures	Main Results
*Drug Alone*						
Saruhashi et al., 2002	Rats, in vitro compressive injury, *n* = 24	Tandospirone, 10 and 100 µM, single dose, Mode: incubated	Monotherapy	Electrical stimulation recording inducing action potential.	Amplitude change and latency change of the evoked action potential	↑ recovery of mean action potential
Landry et al., 2006	Mice, Th9/Th10 transection, 7 day post-Tx, *n* = 36	8-OH-DPAT, 1mg/kg, single dose, Mode: ip	Monotherapy	Quantitative kinematic analysis of hindlimb movements. In situ hybridization	Locomotor-like movements (LM) and non-locomotor-like (NLM) movements. Footstep amplitude. Angular excursion for the ankle.	Induction of locomotor like movement.
Lapointe et al., 2008	Mice, Th9/Th10 transection, 6 days post-Tx, *n* = 40	8-OH-DPAT, 0.5 mg/kg, single dose, Mode: ip ^1^	Monotherapy	Assessment of hindlimb movements	LM, NLM ^2^	Acute LM induction
Jeffrey-Gauthier et al., 2018	Mice, Th7-Th8 hemisection, 2 days post-Tx, *n* = 19	Buspirone, 8 mg/kg/day, single dose, Mode: ip	Monotherapy	Assessment of locomotor recovery. Histology	Number of steps. Number of consecutive steps.	↑ Number of steps taken, Improved paw positioning, ↑ Locomotor spinal networks activity
Develle et al., 2020	Mice, decerebrated, *n* = 20	Buspirone, 8mg/kg, single dose, Mode: ip	Monotherapy	H-reflex recording	Frequency-dependent depression of the H-reflex	↑ Reflex excitability
Ahmed et al., 2021	Rats, C4 bilateral crush injury, 1 week post-injury, *n* = 17	Buspirone, 1 dose/day, 1.5–2.5–3.5 mg/kg, single dose, Mode: ip	Monotherapy	EMG, grip strength, reaching and grasping tasks	Reaching (R) and ^3^ grasping (G) functions	Low doses facilitate R and G and improve forelimb grip strength
Jin et al., 2021	Rats, C4 bilateral crush injury, 1 to 8 weeks post-injury, *n* = 29	Buspirone, 1–2 mg/kg, Single dose Mode: ip	Monotherapy		R and G success rates	Forelimb motor function recovery, performance within 2 weeks of buspirone withdrawal
*Combined with other pharmacology*						
Guertin et al., 2010	Mice, Th9/Th10 transection, 7 days post-Tx *n* = 22	Buspirone, differential 0.5, 1.5, 7.5, or 10 mg/kg, Mode: oral	Tritherapy ^4^	Behavioral assays	Locomotor-like behavior Movement frequency	Locomotor activity induction
Guertin et al., 2012	Mice, Th9/TH10 transection, 1week post-Tx, *n* = 21	Buspirone, 1.5 mg/kg, repetitive administration, Mode: subcutaneous	Tritherapy	Behavioral assays, Histology	Hindlimb movement, induced stepping	Central pattern generator activation, induced episodes of weight-bearing stepping
Slawinska et al., 2012	Rats, Th9/Th10 transection, 10 weeks post-Tx, *n* = 15	8-OH-DPAT, 0.2–0.4 mg/kg, single dose Mode: ip	Quipazine	EMG, Evaluation of hindlimb, locomotor performance	Hindlimb locomotor movements, Weight supported stepping, EMG ^5^	LM activity improvement mitigated the need to activate the LM with exteroceptive stimulation
Radhakrishna et al., 2017	Patients with complete and incomplete SCI, > 3 months post-SCI, *n* = 45	BuspironeGrp1: 10mg, Grp2: 15 mg, Grp3: 25 mg, Grp4: 35 mg, Grp5: 50 mg, Grp6: 75 mg, repetitive administration, Mode: oral	Tritherapy ^6^	Dose-escalation study of Buspirone	EMG, leg movement	↑ EMG activity ↑ locomotor-like characteristics
*Combined with exercise training*						
Ung et al., 2012	Mice, Th9/Th10 transection, 1-week post-Tx, *n* = 43	Buspirone (3 mg/kg)3 times/w × 8 weeks, repetitive administration, Mode: ip	Tritherapy^4^ + training + clenbuterol	LM ^7^ assessment, Muscle fiber immunofluorescence	Locomotor movement, Movement frequency, Cross-sectional area of muscle fibers	↑ locomotor movement and muscle properties, ↑ type II fiber cross sectional area values, ↓ fiber type conversion
Ganzer et al., 2018	Rats, Th8/Th9 transection, 1 week and 8 weeks post-Tx, *n* = 41	8-OH-DPAT, single dose: 0.075–0.125 mg/kg, repetitive administration: 0.125 mg/kg, 5 days/week × 2–8 weeks, Mode: ip	Bike therapy	Acute dose response test, Behavioral testing, Locomotor Scoring Immunohistochemisty	Open field scores, Spinal 5-HT_1A_R densities caudal to the SCI, MAP2 dendrite density	Significant open-field weigh-supported stepping, mediated in part by restoring spinal dendritic density
Jeffrey-Gauthier et al., 2018	Mice, Th7/Th8 hemisection, 2 days post-Tx, *n* = 19	Buspirone, 8 mg/kg/day, single dose, Mode: ip	Treadmill	Assessment oflocomotor recovery, Histology	Number of steps, Step occurrence, Number of consecutive steps	↑ Number of steps taken Improved paw positioning, ↑ Locomotor spinal networks activity
Morgan et al., 2021	Patients with acute traumatic SCI, cohort study from 2011 to 2017, *n* = 84	Inpatient rehabilitation +/− treated by Buspirone	Inpatient rehabilitation +/− treated by Buspirone	Functional scores comparison	Upper extremity motor score, lower extremity motor score, American Spinal Injury Association Impairment Scale, neurological level of injury, and functional impairment measure	↑ 1-year conversion rate to incomplete injury (42.9% with Buspirone vs 21.2% without Buspirone, though this was not significantly different from non-buspirone local controls
Vivodtzev et al., 2021	Patients with complete and incomplete SCI at Th3 or above, <2 years post injury, *n* = 21	Buspirone, 29 ± 17 mg/day × 3 months, repetitive administration, Mode: oral	FES ^8^-rowing	FES-rowing testCardiopulmonaryfunction testing Spirometry	Peak Power output, VO_2_, VCO_2_, Vt, B*f* ^9^	↑ Aerobic capacity ↑ Ventilatory capacity
*Combined with Spinal stimulation*						
Monshonkina et al., 2017	Patients with complete thoracic SCI, > 1-year post-SCI, *n* = 10	Buspirone, Differential-dose 7.5 mg × 2/day × 18 days, repetitive administration, Mode: oral	Spinal stimulation	Percutaneous electrical stimulation of the spinal cord	Activity of the knee and Achille assessment, Rehabilitation status assessment	Muscular force improvement, potentiated the effect of spinal cord stimulation, ↑ pain sensitivity
Gad et al., 2017	Patient with complete SCI at Th9 and L1, 4 years post-SCI, *n* = 1	Buspirone repetitive administration/1 week Mode: oral	Spinal stimulation	EKSO bionics exoskeleton, Painless cutaneous (pcEmc) and Pharmacological (fEmc) enabling motor control EMG	Evoked potential, Robotic assistance, Percent effort, EMG amplitude	↓ robotic assistance, ↑ EMG activity Change in knee angle
Freyvert, Sci Rep. 2018	Patient with complete SCI at C5 or above, >1-year post-SCI, *n* = 6	Buspirone, 15 mg/day × 15 days, repetitive administration, Mode: oral	Spinal stimulation	Handgrip force measurement, EMG	Functional metrics, EMG amplitude, Changes in mean grip strength	↑ Hand function↑ EMG amplitudes

^1^ ip, intraperitoneal; ^2^ LM, Locomotor-like movements and NLM, non-locomotor-like movements; 3 R, Reaching; G, grasping; ^4^ Tritherapy is a combination of Buspirone, L-Dopa, and Carbidopa; ^5^ EMG, electromygram; ^6^ Tritherapy is a combination of Buspirone, L-Dopa, and Carbidopa; ^7^ LM, Locomotor-like movements; ^8^ FES, functional electrical stimulation; ^9^ VO_2_, oxygen consumption; VCO_2_, CO_2_ production; Vt, tidal volume; B*f*, Breathing frequency.

**Table 2 pharmaceuticals-15-00460-t002:** Respiratory effects of 5-HT_1A_ agonists in animals and humans.

Authors	Species/Type of SCI	Drug & Dose	Combined with	Method	Main Outcome Measures	Main Results
Teng et al., 2003	Rats T8 hemicompression 24 h and 7 days post-injury *n* = 16	Buspirone, 1.5 mg/kg, Single dose Mode: ip	Monotherapy	Plethysmography at ambient air and 7% CO_2_ exposure	B*f**,* V_T_ ^1^, Ventilatory response to 7% CO_2_	Normalized B*f*, Vt and respiratory response to 7% CO_2_
Choi et al., 2005	Rats C5 Hemicontusion 2, 4, and 6 weeks post-injury*n* = 44	8-OH-DPAT ^2^ 250 µg/kg, or Buspirone 1.5 mg/kg, Single dose Mode: ip	Monotherapy	Plethysmography at ambient air and 7% CO_2_ exposure Histology	B*f**,* V_T_, Ve while breathing ambient air or 7% CO_2_	↑ hypercapnic ventilatory response to CO_2_, 2 and 4 weeks post injury for up to 4 h
Zimmer et al., 2006	Rats C2 Hemi-transected 24 h/48 h and 1 week post-injury *n* = 52	8-OH-DPAT, 17 µg/kg, Mode: intravenous	Monotherapy	Phrenic activity recording	PO_2_, PCO_2_, Apneic thresholds, Phrenic nerve activity	↑ phrenic activity and amplitude, ↑ phrenic amplitude, ↑ respiratory response and *f* after systemic injection
Maresh et al., 2020	Chronic SCI patients C5-T3 *n* = 8	Buspirone, 30 mg/day × 13 days, repetitive administration, Mode: oral	Monotherapy	Pneumotachometer connected to a tight-fitting nasal mask, ECG	CO_2_ reserve, Apneic threshold Vt, PETCO_2_, PETO_2_ 3	Widened CO_2_ reserve ↓ hypocapnic central sleep apnea
Vivodtzev et al., 2021	Subacute SCI patients < 2 years post injury C5-T3 *n* = 21	Buspirone, 29 ± 17 mg/day × 3 months, repetitive administration, Mode: oral	FES-rowing ^4^	FES-rowing test, Cardiopulmonary function testing, Spirometry	Peak VO_2_, VCO_2_, Vt, B*f* Respiratoy function (FEV_1_ and FVC) ^5^	↑ Aerobic capacity ↑ Ventilatory capacity Changes in ventilatory capacity proportional to changes in respiratory function

^1^ _B*f*_, Breathing frequency; Vt = tidal volume; ^2^ 8-OH-DPAT = 8-hydroxy-2-(di-n-propylamino)tetralin; ^3^ End-tidal Pressure in O_2_ (PETO_2_), End-tidal Pressure in CO_2_ (PETCO_2_); ^4^ Functional electrical stimulation; ^5^ FEV_1_, Forced expiratory volume in one second; FVC, Forced vital capacity.

## Data Availability

Not applicable.

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
