# Peer review of "Serotonin 1A Receptor Pharmacotherapy and Neuroplasticity in Spinal Cord Injury"

_pharmaceuticals, 2022, doi:10.3390/ph15040460_

Round 1

Reviewer 1 Report

The current manuscript titled, Serotonin 1A receptor pharmacotherapy and neuroplasticity in spinal cord injury: a qualitative review deals with the detailed investigation of Serotonin 1A receptor. Since serotoninergic (5-HT) 1A receptor agonists appear as pharmacologic neuromodulators, thus it is implicated to induce functional improvements by increasing activation of spared motoneurons. Despite a good effort, this manuscript lacks presentation. Following suggestions could be added during revision:

  1. What is the meaning of a qualitative review? It may be omitted or replaced.
  2. The introduction is too ambiguous. Please provide a structured ending statement in the introduction section that why this review article is important considering the field, its significance and novelty. What makes this review article unique?
  3. It is essential to identify the research problem that the author has in his mind and state it clearly as a statement of the problem. This will allow readers to travel with the author smoothly.
  4. First, what is the purpose of the study? Then comes who the target audience is? This should be made clear. Second, what is the problem? Moreover, why that problem is significant?
  5. What will be the benefit of the study? The introduction should specifically indicate the significance of the study as a separate paragraph. Moreover, this will inspire readers to take an interest in the subject matter.
  6. Is the study unique, and there is no earlier study or a similar review published? If yes, it should be justified by surveying the relevant literature on the subject.
  7. Is the study comparative? If yes, it must be stated explicitly.
  8. The author should state the methodology applied in the current study. This will help the audience to comprehend the discourse.
  9. Many abbreviations are not described at first appearance. Author should recheck.
  10. Several old or obsolete references are cited. I suggest to reduce number of references. Formatting errors also seen.
  11. Perspectives and future directions should be elaborated.

Author Response

Response to the reviewers

Pharmaceuticals-#1591306:

We thank the reviewers for their positive statement on our work and their valuable suggestions. The following is a point by point response to the comments along with the location of the resulting changes to the manuscript.

Reviewer 1:

C1: The current manuscript titled, Serotonin 1A receptor pharmacotherapy and neuroplasticity in spinal cord injury: a qualitative review deals with the detailed investigation of Serotonin 1A receptor. Since serotoninergic (5-HT) 1A receptor agonists appear as pharmacologic neuromodulators, thus it is implicated to induce functional improvements by increasing activation of spared motoneurons. Despite a good effort, this manuscript lacks presentation. Following suggestions could be added during revision:

R1: We thank the reviewer for his/her time and effort to review the paper and his/her fruitful suggestions.

C2: What is the meaning of a qualitative review? It may be omitted or replaced.

R2: We initially wanted to distinguish our work to systematic review and meta-analyze and but we agree to omit it.

C3: The introduction is too ambiguous. Please provide a structured ending statement in the introduction section that why this review article is important considering the field, its significance, and novelty. What makes this review article unique?

R3: We are now providing a clear and structured ending statement in the introduction section (page 7).

C4: It is essential to identify the research problem that the author has in his mind and state it clearly as a statement of the problem. This will allow readers to travel with the author smoothly.

R4: We agree with the reviewer and we have now clearly stated the research problem in the introduction section (page 7).

Paragraph added page 7: In this unique review, we hence discuss previous and emerging evidences showing the value of 5HT1A receptor agonist therapy for motor and respiratory limitations in SCI. More precisely, the purpose of our study is to provide a robust insight of the work delivered in the last two decades on how 5HT1A receptor agonists can be beneficial in clinical cases outside of their common use in psychiatry as anxiolytics. Particularly, studies suggest that 5HT1A receptor agonist therapies led to improvement in both locomotor and respiratory functions, and that this effect was amplified when 5HT1A receptor agonist treatment were combined with other drugs, electrical stimulation of the spinal cord, or exercise training (rehabilitation). This is the first study providing an exhaustive overview of the literature on the effect of 5HT1A on locomotor/respiratory neuroplasticity in SCI, not only providing new insight on the use of this pharmacotherapy in rehabilitation of individuals with SCI for clinicians, but also future directions for scientist aiming at developing research in neuroplasticity in SCI.”

C5: First, what is the purpose of the study? Then comes who the target audience is? This should be made clear. Second, what is the problem? Moreover, why that problem is significant?

R5: We have now clearly presented the purpose of our study, the target audience and the significance of the study (page 7).

C6: What will be the benefit of the study? The introduction should specifically indicate the significance of the study as a separate paragraph. Moreover, this will inspire readers to take an interest in the subject matter.

R6:  Please see response to C5

C7: Is the study unique, and there is no earlier study or a similar review published? If yes, it should be justified by surveying the relevant literature on the subject.

R7: Indeed, this study is unique, we have now clearly mentioned it in the introduction page 7.

C8: Is the study comparative? If yes, it must be stated explicitly.

R8: We did not compare quantitatively the results of the studies but we highlighted their similarities and differences.  

C9: The author should state the methodology applied in the current study. This will help the audience to comprehend the discourse.

R9: The methods are now stated in the manuscript at the end of the introduction section page 7-8.

“Herein, we searched for published studies investigating Serotonin 1A receptor pharmacotherapy and neuroplasticity in spinal cord injury. PubMed and Web of Science databases were screened using the following key words and MeSH terms: [spinal cord injury] AND [Serotonin 1A receptor] AND [locomotor] OR [respiration]. Original studies in animal or in human that met the following criteria were included: i) study design: drug alone or combined; within or between groups comparison, non-randomized between-groups studies, randomized controlled studies, cross-sectional studies, case and cohort studies (for human); ii) subjects: SCI models including total or partial transection, contusion, decerebration / individuals with spinal cord injury and iii) outcomes: any outcomes related to locomotor or respiratory function. Non–English language articles, review articles and congress abstracts were excluded.” 

C10: Many abbreviations are not described at first appearance. Author should recheck.

R10: We apologize for these omissions and we have now checked for all abbreviations in the manuscript.

C11: Several old or obsolete references are cited. I suggest reducing number of references. Formatting errors also seen.

R11: We have now checked for all references and reduced their number when possible.

C12: Perspectives and future directions should be elaborated.

R12: We have now elaborated our paragraph of future direction page 30.

Reviewer 2 Report

Good review.

 Minor English edits

Author Response

Response to the reviewers

Pharmaceuticals-#1591306:

We thank the reviewers for their positive statement on our work and their valuable suggestions. The following is a point by point response to the comments along with the location of the resulting changes to the manuscript.

Reviewer 2:

C1: Good review.

Minor English edits

R1: We thank the reviewer for his/her time and effort to review the paper and we are grateful for his very positive comment. We have carefully checked the manuscript and hope to have reduce spelling errors.

Reviewer 3 Report

I think the review is already well-written and it can be published in present form.

Author Response

Response to the reviewers

Pharmaceuticals-#1591306:

We thank the reviewers for their positive statement on our work and their valuable suggestions. The following is a point by point response to the comments along with the location of the resulting changes to the manuscript.

Reviewer 3:

C1: I think the review is already well-written and it can be published in present form.

C1: Thank you!

Reviewer 4 Report

In the present paper,  previous and emerging evidence of the role played by 5HT1A receptor agonist therapies for motor and respiratory limitations in SCI are discussed. As from them, authors propose mechanistic hypotheses and potential clinical impact for 5-HT1A agonists to induce neuroplasticity and locomotor and respiratory recovery after SCI.  

No ethical concerns.

The introduction is succinct, but clearly explains the causes of respiratory and locomotor function loss after high cervical spinal cord injury, the relationship between the two deficits and the role of 5HT1As as pharmacological neuromodulators promoting sprouting after SCI.

The articles selected as sources are relevant and the information they provide is sufficiently well presented and discussed. With regard to the combinatorial effects of 5-HT1A agonist with rehabilitation or spinal cord stimulation, the authors refer to both therapies in a very generic way, which is understandable and appropriate for this article and the journal to which it is submitted. However, a sentence should be added to alert the reader who is not an expert in rehabilitation and/or spinal cord stimulation to the fact that there are different therapeutic programmes in both cases and that the effects of the different programmes are different. However, a sentence should be added to alert the non-expert reader in rehabilitation and/or spinal cord stimulation that the therapeutic methods and programmes in both cases are different and that the effects of the different programmes may not be exactly superimposable.

Figures and table adequately support and clarify the text.

Mechanistic hypothesis, clinical impact and future perspectives are concrete and very interesting.

Author Response

Response to the reviewers

Pharmaceuticals-#1591306:

We thank the reviewers for their positive statement on our work and their valuable suggestions. The following is a point by point response to the comments along with the location of the resulting changes to the manuscript.

Reviewer 4:

Comments and Suggestions for Authors

In the present paper, previous and emerging evidence of the role played by 5HT1A receptor agonist therapies for motor and respiratory limitations in SCI are discussed. As from them, authors propose mechanistic hypotheses and potential clinical impact for 5-HT1A agonists to induce neuroplasticity and locomotor and respiratory recovery after SCI.  

No ethical concerns.

The introduction is succinct, but clearly explains the causes of respiratory and locomotor function loss after high cervical spinal cord injury, the relationship between the two deficits and the role of 5HT1As as pharmacological neuromodulators promoting sprouting after SCI.

The articles selected as sources are relevant and the information they provide is sufficiently well presented and discussed. With regard to the combinatorial effects of 5-HT1A agonist with rehabilitation or spinal cord stimulation, the authors refer to both therapies in a very generic way, which is understandable and appropriate for this article and the journal to which it is submitted.

However, a sentence should be added to alert the reader who is not an expert in rehabilitation and/or spinal cord stimulation to the fact that there are different therapeutic programmes in both cases and that the effects of the different programmes are different.

R1: We are very grateful to the reviewer for his/her positive comments on our manuscript. We agree with this suggestion and we have now included a sentence specifying the difference between the 2 therapeutic strategies page 19 & 20.

Of note, spinal stimulation and rehabilitation are two different programs, triggering different effects. Indeed, spinal cord stimulation is a local procedure consisting in an electrical current application via cutaneous electrodes on the spinal cord to activate intraspinal neural networks [41], while during rehabilitation, individuals with SCI perform exercise with their non-paralyzed muscles and/or their paralyzed muscles using devices such as electrical stimulation, weight-bearing system and/ or exoskeleton, therefore triggering neuromuscular or muscular effects [6].

C2: Figures and table adequately support and clarify the text. Mechanistic hypothesis, clinical impact and future perspectives are concrete and very interesting.

R2: Thank you!

Round 2

Reviewer 1 Report

Accept